# Audiobook synthesis with long-form neural text-to-speech

*Weicheng Zhang, Cheng-Chieh Yeh, Will Beckman, Tuomo Raitio, Ramya Rasipuram,*
*Ladan Golipour, David Winarsky*

Apple, USA

## Abstract

Despite recent advances in text-to-speech (TTS) technology, auto-narration of long-form content such as books remains a challenge. The goal of this work is to enhance neural TTS to be suitable for long-form content such as audiobooks. In addition to high quality, we aim to provide a compelling and engaging listening experience with expressivity that spans beyond a single sentence to a paragraph level so that the user can not only follow the story but also enjoy listening to it. Towards that goal, we made four enhancements to our baseline TTS system: incorporation of BERT embeddings, explicit prosody prediction from text, long-context modeling over multiple sentences, and pre-training on long-form data. We propose an evaluation framework tailored to long-form content that evaluates the synthesis on segments spanning multiple paragraphs and focuses on elements such as comprehension, ease of listening, ability to keep attention, and enjoyment. The evaluation results show that the proposed approach outperforms the baseline on all evaluated metrics, with an absolute 0.47 MOS gain in overall quality. Ablation studies further confirm the effectiveness of the proposed enhancements.

**Index Terms**: speech synthesis, prosody, natural language processing, audiobook synthesis

## 1. Introduction

Recent advances in text-to-speech (TTS) synthesis, such as Tacotron [1], FastSpeech 2 [2], and WaveRNN [3], have enabled neural network-based TTS systems. A typical neural TTS system consists of two main neural networks: an acoustic model and a vocoder. The acoustic model takes graphemes or phonemes as input and predicts Mel-spectrogram as an intermediate feature, while the vocoder takes Mel-spectrogram as input and generates speech samples. Neural TTS can produce quality that is close to natural speech [4, 5], and synthesis applications like virtual agents can now have human-like prosody. However, the scope of the input text and the prosodic diversity of the output are fairly constrained in such applications. The input text is frequently made up of single, short sentences, and the vocabulary is small and often seen in the training data. Finally, the required prosodic variations mostly encompass simple declarative and interrogative sentences.

The development of synthetic audiobooks deals with a broad range of content. There are differences in sentence length, from short to long sentences; in sentence style, from simple to compound to complex sentences; in sentence types, from declarative to interrogative to exclamatory and more; there are other elements in books such as dialogue, and finally the speaking style and expressivity of speech is dependent on the content and the context. As a result, the TTS system must be able to cover the prosodic range required for a variety of content while ensuring that the generated speech is not only intelligible but also enjoyable to listen to. We specifically add the following four enhancements to improve long-form reading required in audiobooks synthesis:

1. We incorporate semantic and syntactic information by integrating BERT [6] embeddings learned from large amounts of unlabeled text data. BERT embeddings contain information about the semantics of the phrase and the relevance of each word, thus producing more natural prosody [7, 8, 9, 10].

2. While synthesizing long-form expressive text, we need to synthesize a wide range of speaking styles, and the speaking style needs to be suitable for the content. We integrate style embeddings [11, 12, 13] to our baseline neural TTS system, where the style embeddings are predicted from text.

3. Conventional TTS systems process each sentence separately. As a result, they fail to capture inter-sentence prosodic dependencies. In this work, we increase the input and output to the model to encompass multiple sentences at a time to learn high-level prosody within the paragraph and to transition smoothly between sentences and phrases.

4. To learn the pronunciation of a large vocabulary and diverse prosody required in audiobook synthesis, we pre-train our models on long-form data.

### 1.1. Relation to prior work

Public domain audiobooks have been widely used as source material to train TTS models. For example, LibriSpeech [14] and a derivative dataset LibriTTS [15] are commonly used as they provide large amounts of relatively high-quality speech material. The Blizzard Challenge, an annual event for evaluating TTS systems, has included audiobooks as their training data and the target for evaluation [16, 17]. Since then, some studies have addressed various aspects in the synthesis of audiobooks [18, 9, 19, 20, 21, 22, 23, 24]. The closest works to ours are presented in [9, 22, 23, 24], where similar techniques are proposed to improve multi-sentence or audiobook TTS. In this work, we propose a TTS system to synthesize long-form context like audiobooks using long-form neural text-to-speech, and we evaluate the synthesis on segments spanning multiple paragraphs. Previous work on evaluating long-form TTS has been conducted in [25] where single-sentence evaluation was concluded to be insufficient. An evaluation protocol for evaluating TTS for audiobook reading was proposed in [26], inspired by the rating scales in [27]. In this work, we present an evaluation method with nine questions specifically aimed for evaluating the listening experience of synthetic audiobooks.

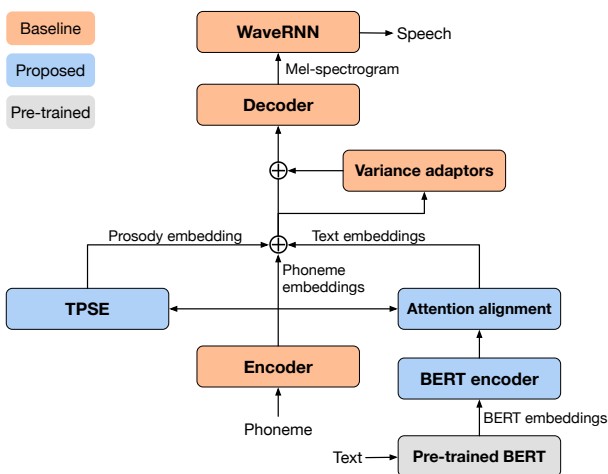

Figure 1: *Neural TTS architecture with proposed improvements.*

## 2. Methods

The proposed system has two main modules: an acoustic model, similar to FastSpeech 2 [2], that takes phonemes and punctuations as input and generates a Mel-spectrogram, and a vocoder, similar to WaveRNN [3], that generates speech samples conditioned on the Mel-spectrogram. In this work, we focus on the acoustic model and explain the enhancements for generating high quality audiobooks, since the pronunciation and prosody are mostly determined by the Mel-spectrogram. For the detailed implementation of the vocoder, we refer to [28]. In this section, we first introduce the baseline acoustic model architecture, and then describe additional modeling and training data enhancements that enable high-quality audiobook synthesis.

### 2.1. Acoustic model architecture

**Baseline.** Our baseline acoustic model, shown in Fig. 1, follows the FastSpeech 2 architecture [2] with certain modifications. Input to the system is phoneme sequence along with punctuation and word boundaries, and the output is Mel-spectrogram. The model is based on a feed-forward Transformer (FFT) [29, 30] encoder and dilated convolution decoder. The encoder consists of an embedding layer that converts the phoneme sequence to a dense representation and appends positional encodings. This is followed by a series of FFT blocks where each FFT block consists of a self-attention layer [29] and 1-D convolution layers along with layer normalization and dropout to output phoneme embeddings. The phoneme embeddings are fed to the variance adaptors that predict phone-wise duration, pitch, and energy. The variance adaptors consist of 1-D convolution layers, layer normalization, and dropout. Instead of using pitch spectrograms as in [2], we use continuous pitch with quantization and a projection to an embedding. The predicted pitch and energy are then added to the phoneme embeddings and upsampled according to the predicted phone-wise durations. The decoder consists of a series of dilated convolution stacks instead of the original FFT blocks as in [2], which improves model inference speed and memory usage. Finally, the decoder converts the upsampled encoder sequence into a Mel-spectrogram.

**BERT.** The input to the baseline model is a phoneme sequence, which contains pronunciation information but is lacking semantic information. Moreover, the limited TTS training data does not help in learning higher-level generalizations for prosody.

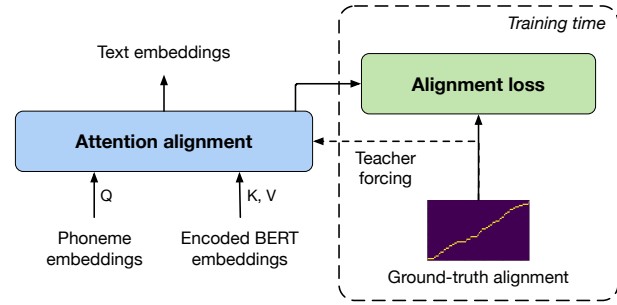

Figure 2: *Proposed method for learning the alignment between BERT tokens and phonemes.*

Therefore, the acoustic model has a limited capability to infer appropriate prosody for various contexts in diverse speech material. We incorporate semantic and syntactic information through BERT [6] representations learned from large amounts of text data to improve prosody. BERT is a multi-layer bidirectional Transformer encoder that has been pre-trained on large amounts of unlabeled text data through self-supervision. The pre-trained BERT model generates word representations which are then fed to a BERT encoder, which is trained with our system to accommodate the additional input. The BERT encoder is similar to the phoneme encoder but without the positional encoding since that information has been provided inherently through the BERT embeddings themselves. In our experiments, fine-tuning the BERT model with audiobook material did not show clear improvement over the pre-trained model, and therefore we do not fine-tune BERT model in this work.

It is not straightforward to combine BERT and phoneme sequences as BERT represents input text as word pieces which is not aligned with phoneme representation. In this work we propose to align the BERT token sequence and the phoneme sequence using a learned alignment as shown in Fig. 2. First, we use heuristic rules to align BERT tokens and phoneme sequences of the training data. During training, the attention alignment module uses the rule-based ground-truth alignment for teacher forcing. BERT token sequence is upsampled and concatenated with phoneme embeddings. At the same time we use ground-truth alignment as target to compute the alignment loss. During inference, the attention alignment module is used to align BERT tokens and phoneme sequences.

**Text-based prosody prediction.** To further improve the prosody and expressivity of audiobook synthesis, we incorporate a prosody encoder to the baseline architecture. The prosody encoder is based on the Text-Predicted Global Style Token (TP-GST) method, and more specifically on Text Predicted Sentence Embedding (TPSE) [13]. During training, the GST module learns to capture global variation in the training data using a reference encoder, and a style attention module that learns style vectors. Given a Mel-spectrogram, GST module predicts style embedding that is a convex combination of style vectors. The TPSE module takes encoder embeddings as input and learns the relationship between the text and the style embedding. At inference time, style embeddings are predicted given the encoder embeddings using the TPSE module. TP-GST was proposed to work with an auto-regressive Tacotron model [1]. In this work, we use TP-GST with FastSpeech 2 model, and therefore replace recurrent components of the reference encoder and the TPSE module with feed-forward convolution layers. Phoneme encoder embeddings and the style embedding from TPSE together act as input to the decoder and the

variance adaptors so that the style embedding can influence both the Mel-spectrogram and acoustic feature generation.

## 2.2. Data enhancements

**Long-context modeling.** Conventional TTS systems model one utterance at a time during both training and inference. For applications like audiobook synthesis, where the prosody of utterances depends not only on the content but also on the context, utterance-level training and inference does not generalize well for long-form reading.

In this work, we propose long-context modeling where we model multi-sentence input and multi-sentence output to learn intra-sentence prosody and to transition smoothly between sentences and paragraphs. In long-context modeling, the input and output are consecutive sentences either from the same paragraph or different paragraphs of the recording. To learn several contexts, the combination of multi-sentence input/output is done recursively so that utterances appear multiple times in the training data: once as the first utterance, once as the second utterance, etc.

**Pre-training.** Synthesizing long-form material like audiobooks present various challenges such as generalizability to unseen input, necessity to model the pronunciation of a large vocabulary, and diverse prosody based on content and context. To overcome these challenges we first pretrain our models on long-form content. Several methods can be applied to help the model generalize better, such as multi-speaker modeling [31, 32], pre-training and fine-tuning [33, 34, 35], or meta-learning [36]. In this work, we use a simple but effective model pre-training and fine-tuning.

# 3. Experiments

## 3.1. Data

Two proprietary American English single-speaker audiobook datasets were used to test the proposed techniques: a 29-hour dataset of a female voice and a 42-hour dataset of a male voice. For pre-training, we prepared corresponding pre-training datasets for both voices. To pretrain the female voice, we used 308 hours of long-form material from 36 female speakers, and for the male voice, we used 208 hours from 47 male speakers. To train a conventional baseline TTS system that does not include any enhancements proposed in the paper, we used 36 hours of speech from a female voice talent recorded in a consistent style for conventional TTS purposes.

## 3.2. Systems

We used the following systems and samples in evaluation:

- **Baseline-conventional**: Baseline architecture, described in Sec. 2.1, trained with the conventional TTS dataset.

- **Baseline-audiobook**: Baseline architecture trained with the audiobook datasets.

- **Proposed**: Baseline architecture combined with the proposed BERT, TPSE, long-context, and pre-training enhancements, and trained with the audiobook datasets.

- **Natural recordings**: Recordings from audiobooks for comparison, includes both male and female speech samples.

The baseline-conventional system was trained with the conventional TTS dataset, while the baseline-audiobook and proposed systems were trained with the audiobook datasets described in

Sec. 3.1. Baseline-conventional and baseline-audiobook systems are trained on respective datasets for 140k steps using distributed training algorithm SyncSGD [37] on 16 GPUs with a batch size of 32. We chose Adam optimizer ($\beta_1 = 0.9$, $\beta_2 = 0.999$, $\epsilon = 1e-8$) with an initial learning rate of $1e-3$.

To train the proposed system, we build long-context pre-train and fine-tune datasets by combining the input and output of two consecutive sentences (2-sentence modeling). To avoid memory issues, we set a threshold of 200 phonemes for the combined input of the two sentences. The proposed system was first pre-trained on the pre-train dataset for 400k steps using the same settings as in the baseline systems, and then fine-tuned (all the parameters of the model are fine-tuned) on the target dataset with smaller initial learning rate of $1e-4$ for another 140k steps.

## 3.3. Evaluation

TTS systems are typically assessed at sentence-level using mean opinion score (MOS) [38], where listeners listen to a sentence and rate it on a scale of 1 to 5. However, the traditional approach of evaluating at sentence-level and with focus only on overall opinion is not appropriate for evaluating synthetic audiobooks as there are many factors that contribute to the listening experience. Evaluation of long-form synthesis has been investigated for example in [25] where it was shown that the traditional way of evaluating sentences in isolation is not sufficient. In this work, we propose to evaluate synthesized audiobooks using much longer segments (approximately two minutes), and listeners not only rate the overall opinion but rate multiple dimensions on a scale of 1 to 5 as follows:

1. **Overall:** What is your overall opinion of the audiobook sample you just heard? (1−Very poor, 2−Poor, 3−Ok, 4−Good, 5−Excellent)

2. **Personality:** How would you rate this narrator's character or personality? (same rating scale as above)

3. **Voice-content match:** Is this narrator's voice appropriate for the content in the sample? (same rating scale as above)

4. **Attention:** How well was the narrator able to keep your attention? (1−Difficult to maintain attention, 2−Considerable effort required to maintain attention, 3−Moderate effort required to maintain attention, 4−Easy to maintain attention, 5−Very easy to maintain attention)

5. **Dialog distinction:** How easy was it to distinguish narration from character dialog in this sample? (1−Very difficult, 2−Difficult, 3−Neutral, 4−Easy, 5−Very easy, NA−Not applicable)

6. **Ease:** Would it be easy or difficult to listen to this voice for long periods of time? (same rating scale as above)

7. **Comprehension:** How well did you follow this narrator's storytelling? (1−Very difficult to follow, 2−Had some difficulties, 3−Fairly well, 4−Well, 5−Very well)

8. **Pausing:** How would you rate the narrator's pausing in general in this sample? (1−Very poor, missing or misplaced pauses, 2−Poor in most cases, 3−Ok, 4−Good in most cases, 5−Excellent, pauses where expected)

9. **Pacing**: How would you rate the narrator's pacing in general in this sample? (1−Very poor, unnaturally fast or slow, 2−Poor in most cases, 3−Ok, 4−Good in most cases, 5−Excellent, pace as expected)

In the listening test, for each of the two voices, we had 55 two-minute samples for the baseline-audiobook and proposed

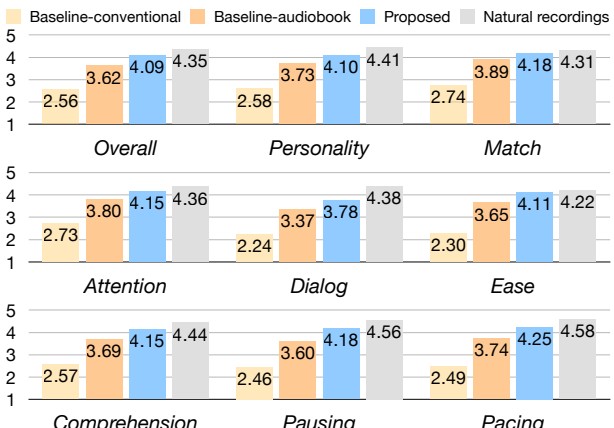

Figure 3: *Results of the MOS evaluation. The scores for male and female voices are averaged to simplify the illustration.*

systems. For the baseline-conventional system and the natural recordings, we had 15 two-minute samples. All source text for the evaluation was long-form content from books not used for training the models. The samples were evaluated by 204 listeners with 15 responses per sample for each of the 9 questions, resulting in a total of $(4\times55+2\times15)\times9\times15$=33,750 ratings.

The results, depicted in Fig. 3, show that the proposed method outperforms both baseline systems for all of the 9 questions. There is an absolute 0.46 MOS gain in overall quality for the proposed system compared to the baseline-audiobook system. Results also show that baseline-audiobook system results in better synthesis compared to the baseline-conventional system, indicating that training with dataset close to the target task improves the synthesis quality. The nine evaluation metrics correlate with each other with *personality* and *ease* being most correlated with the *overall* rating. *Dialog distinction* was least correlated with the *overall* rating and had the lowest score for the proposed system, which suggests room for improvement and future work on this topic.

### 3.4. Ablation studies

We evaluated the effectiveness of the various proposed enhancements using ablation studies. We removed each of the model-based and data-based enhancements individually and together as shown in Table 1, and evaluated the effect on quality in comparison to the full proposed system. We synthesized 110 one-minute long samples for each of the 6 systems for both voices. We received 12 ratings per sample from 372 listeners using the comparative mean opinion score (CMOS) test on a scale of -3 (much worse) to +3 (much better), resulting in a total of $6\times2\times12\times110$=15,840 ratings. We measured significant differences by running Wilcoxon signed-rank test with the null-hypothesis being samples from each side are from the same distribution. We rejected the null-hypothesis if $p < 0.05$.

The results, depicted in Table 1, show that all the proposed enhancements have a significant positive effect on quality with BERT and long-context modeling resulting in higher quality gains compared to pre-training and TPSE. The results indicate that incorporating BERT embeddings, which contain information about the semantics of the phrase and the importance of each word, help the system produce more natural prosody. Also, the long-context modeling helps in learning higher-level prosody inside the paragraph and transition smoothly between

Table 1: *CMOS scores and p-values of the ablation studies.*

| System | Female | | Male | |
|---|---|---|---|---|
| | Score | p-value | Score | p-value |
| Proposed | 0 | N/A | 0 | N/A |
| − TPSE | -0.14 | ≪0.001 | -0.00 | 0.930 |
| − BERT | -0.49 | ≪0.001 | -0.24 | ≪0.001 |
| − TPSE − BERT | -0.47 | ≪0.001 | -0.24 | ≪0.001 |
| − Pre-train | -0.15 | ≪0.001 | -0.07 | 0.034 |
| − Long-context | -0.42 | ≪0.001 | -0.58 | ≪0.001 |
| − Pre-train − long-context | -0.57 | ≪0.001 | -0.41 | ≪0.001 |
| + 6-sentence long-context | +0.18 | ≪0.001 | +0.11 | 0.018 |

sentences. The pre-training brings improvements to both male and female voices, but is less significant in comparison to BERT and long-context methods. We hypothesize that this is due to the model learning and benefiting more from the high quality fine-tune recordings, and hence the effect from using pre-train data is reduced. Improvements from TPSE are clear for the female voice but less clear for male voice. This discrepancy might be due to more varied and consistent prosody in the female data, while the style variation was less prominent in the male data.

### 3.5. Long-context with 2 sentences vs. 6 sentences

Encouraged by the results on the proposed system with 2-sentence modeling, we further increased the input/output from 2 sentences to 6 sentences. As shown in the last row of Table 1, for both male and female voices, long-context modeling with 6 sentences outperforms 2 sentences.

## 4. Conclusions

Synthesizing long-form content like books is challenging for TTS systems due to the wide range of speaking styles, long context dependencies, and a generally high requirement for quality from the listeners. In this paper we proposed four enhancements to the baseline neural TTS system to overcome the challenges, namely: incorporation of BERT embeddings, explicit prosody prediction from text, long-context modeling over multiple sentences, and pre-training on large amounts of data. Another challenging aspect of long-form content is the evaluation, as the assessment should not only focus on quality and expressivity but also on experience and enjoyment. Towards that we evaluated long-form synthesis of audiobooks from different perspectives, such as enjoyment, comprehension, ability to keep attention, and ease of listening, on segments spanning multiple paragraphs. The evaluation results show that the proposed approach outperforms the baseline in all evaluated dimensions, with an absolute 0.46 MOS gain in overall quality. Furthermore, the proposed enhancements reduce the huge gap between baseline synthesis and natural recordings for audiobook synthesis. Ablation studies show that BERT and long-context modeling result in significant quality improvements followed by some but less consistent improvements from pre-training and prosody prediction.

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
