# OpenReview forum: "Audiobook synthesis with long-form neural text-to-speech"
_Interspeech.org/2023/Workshop/SSW — SSW12_

### Official Review · Reviewer_vZn4 · 2023-06-02
**Excellent contribution to improved audiobook synthesis**

**Rating:** 8
**Confidence:** 5

**Review:**

This paper about audiobook synthesis using long-form neural TTS is very interesting. The authors show a detailed ablation study that shows the effect of each of the improvements made wrt the baseline system. They also present novel ways to evaluate the output quality, not just relying on MOS of the naturalness but presenting longer fragments of speech (2 minutes) and asking the listeners questions about the narrator's character/personality, the ease of listening, dialog distinction, pausing and pacing.

One thing that they don't mention is the effect of the enhancements (including BERT embeddings, TPSE, and long context) on the complexity/inference speed of the TTS system.

---

> ### Author Response · Authors · 2023-06-23
> **Thank you for the valuable feedback and comments**
>
> Re. "One thing that they don't mention is the effect of the enhancements (including BERT embeddings, TPSE, and long context) on the complexity/inference speed of the TTS system." The added components do add complexity and make the inference slower. The performance optimization is left as future work.

---

### Official Review · Reviewer_LL8K · 2023-06-09
**This paper is focusing on long form generation for TTS. Compared to the baseline FastSpeech2, several methods have been proposed including: using BERT embedding, Txt predicted prosody control, long content modelling and pretraining on dataset. The paper is very easy to read. However, the novelty of the paper is too limited.**

**Rating:** 5
**Confidence:** 5

**Review:**

For BERT embedding and pretrain on dataset, this is almost a standard process, it is hard to say those are the new features added in the system. Also, For text prediction of prosody, it uses the exact method used in the paper “Predicting Expressive Speaking Style From Text In End-To-End Speech Synthesis”. For the key part of using multiple sentences in the training, the paper did not mention much. So it is difficult to evaluate the contribution of the paper.
For the long form, please give a more detailed definition. Whether it is more about the sentence length or the speaking style (e.g: http://wikipedia.org/).
For audiobooks, there are some super expressive voices. Can authors mention whether this has been tested or not. For the testing sentences, is it a more plain reading style similar to assistant voice or different genres?
Similar to point 1, the paper should focus on more of the long form. How it is modelled, and how the long sentences or multiple sentences are processed. However, this part is only briefly mentioned in 3.3.

---

> ### Author Response · Authors · 2023-06-23
> **Thank you for the valuable feedback and comments**
>
> Re. "For the long form, please give a more detailed definition. Whether it is more about the sentence length or the speaking style (e.g: http://wikipedia.org/). Long-form in this work denotes audiobook content. It consist both long and short sentences, but the main point is that the sentences in sequence are meaningfully correlated with each other. The content is narrated with an appropriate prosody for the long-form audiobook content.
>
> Re. "For audiobooks, there are some super expressive voices. Can authors mention whether this has been tested or not." The audiobook material used in the work consists of fictional books narrated in the style appropriate for the genre and includes expressive parts as well.
>
> Re. "For the testing sentences, is it a more plain reading style similar to assistant voice or different genres?" All testing sentences were also long-form audiobook content. We have clarified this in the final version of the paper.

---

### Decision · Program_Chairs · 2023-06-14

**Decision:**

Accept

**Comment:**

SSW2003 received 45 papers. The acceptance rate is 82%. We are pleased to inform you that your paper has been accepted by the SSW2023 Program Committee. Please read the reviews carefully and submit your camera-ready paper by June 28th. Most reviewers performed a detailed review. Please answer to their questions and consider their comments. Note that camera-ready papers are credited with one extra page to allow authors to consider reviewers’ suggestions. So max 7 pages in total including figures & refs.
The deadline for submitting the revised version (with full non-anonymized authors and refs!) is 28th June.